# Remineralization rate of terrestrial DOC as inferred from $CO_2$ supersaturated coastal waters

Filippa Fransner[1,*], Agneta Fransson[2], Christoph Humborg[3,4], Erik Gustafsson[3], Letizia Tedesco[4], Robinson Hordoir[6], and Jonas Nycander[1]

[1]Department of Meteorology and Bolin Centre for Climate Research, Stockholm University, Stockholm, Sweden.
[2]Norwegian Polar Institute, Fram Centre, Tromsø, Norway.
[3]Baltic Nest Institute, Baltic Sea Centre, Stockholm University, Stockholm, Sweden.
[4]Faculty of Biological and Environmental Sciences, Tvärminne Zoological Station, University of Helsinki, Hanko, Finland.
[5]Finnish Environment Institute, Marine Research Centre, Helsinki, Finland.
[6]Institute of Marine Research and Bjerknes Centre for Climate Research, Bergen, Norway
[*]now at Geophysical Institute, Bergen University and Bjerknes Centre for Climate Research, Bergen, Norway

**Correspondence:** Filippa Fransner (filippa.fransner@hotmail.se)

**Abstract.** Coastal seas receive large amounts of terrestrially derived organic carbon (OC). The fate of this carbon, and its impact on the marine environment, is however poorly understood. Here we combine underway $CO_2$ partial pressure ($pCO_2$) measurements with coupled 3D hydrodynamical-biogeochemical modelling to investigate whether remineralization of terrestrial dissolved organic carbon (tDOC) can explain $CO_2$ supersaturated surface waters in the Gulf of Bothnia, a subarctic estuary. We find that a substantial remineralization of tDOC, and that a strong tDOC induced light attenuation dampening the primary production, is required to reproduce the observed $CO_2$ supersaturated waters in the nearshore areas. A removal rate of tDOC of the order of one year, estimated in a previous modelling study in the same area, gives a good agreement between modelled and observed $pCO_2$. The remineralization rate is on the same order as bacterial degradation rates calculated from published incubation experiments, suggesting that bacteria has the potential to cause this degradation. Furthermore, the observed high $pCO_2$ values during the ice-covered season argues against photochemical degradation as the main removal mechanism. All of the remineralized tDOC is outgassed to the atmosphere in the model, turning the northernmost part of the Gulf of Bothnia to a source of $CO_2$ to the atmosphere.

*Copyright statement.* TEXT

## 1 Introduction

Rivers bring large amounts of organic carbon to the coastal seas, linking the terrestrial and oceanic carbon cycles. The riverine organic carbon influences the carbon cycling in coastal seas by providing an external carbon source for bacteria, as well as hampering the primary production by making the coastal waters more turbid (Hessen et al., 2010; Wikner and Andersson, 2012; Bauer et al., 2013). The fate of tDOC in coastal and oceanic waters, and to what extent it undergoes remineralization

by bacteria and photochemical processes, is however poorly constrained (Blair and Aller, 2012). Whereas conservative mixing of tDOC with salinity (Mantoura and Woodward, 1983; Dittmar and Kattner, 2003) points towards an inert behaviour, other studies suggest that there is a large removal, mainly by bacterial and photochemical degradation (Benner and Kaiser, 2011; Fichot and Benner, 2014). The high $pCO_2$ measured in many inner estuaries (Frankignoulle et al., 1998; Borges et al., 2005; Anderson et al., 2009) further indicates that a substantial remineralization of tDOC could take place, but it is not clear how much of this signal is caused by lateral transport of $CO_2$ oversaturated waters from rivers and wetlands (Raymond et al., 2000; Cai, 2011).

The Gulf of Bothnia, in the Northern Baltic Sea (Figure 1), is a subarctic estuary that receives large amounts of allochthonous organic carbon (Sandberg et al., 2004; Alling et al., 2008; Deutsch et al., 2012; Hoikkala et al., 2015) originating from surrounding coniferous forests and peatlands. Recent isotope and modelling studies have shown that a majority of this terrestrially derived organic carbon is removed in the transit from estuarine to more oceanic waters (Alling et al., 2008; Deutsch et al., 2012; Gustafsson et al., 2014; Fransner et al., 2016; Seidel et al., 2017), but no direct evidence of the responsible processe(s) exists, and the time scales of the removal are unclear (Fransner et al., 2016). Upscaling of small scale experiments in the Baltic Sea suggests that photochemical remineralization could account for a major removal (Aarnos et al., 2012), while only a small fraction is available for bacterial degradation (Wikner et al., 1999; Asmala et al., 2013, 2014a; Herlemann et al., 2014; Figueroa et al., 2016; Kuliński et al., 2016) and flocculation processes (Asmala et al., 2014b). Other studies, showing that phytoplankton production of organic carbon is not large enough to support the secondary production, suggest on the other hand that the bacterial production to a large degree is supported by tDOC (Zweifel et al., 1995; Kuparinen et al., 1996; Sandberg et al., 2004). Based on observed $pCO_2$ values, mainly from offshore waters, Löffler et al. (2012) calculated that the Bothnian Bay is a slightly heterotrophic system. Whether this net heterotrophy is due to discharge of river waters supersaturated in $CO_2$, or remineralization of tDOC into dissolved inorganic carbon (DIC), remains to be investigated. To better understand the dynamics of tDOC, observations are needed in the nearshore areas, where the largest tDOC concentrations and likely also the largest tDOC removal takes place (Deutsch et al., 2012).

Here we explore the remineralization dynamics of terrestrial dissolved organic carbon in the Gulf of Bothnia by combining high resolution underway $pCO_2$ measurements with numerical simulations from a 3D coupled hydrodynamic-biogeochemical model. The underway $pCO_2$ measurements cover $CO_2$ supersaturated nearshore waters next to some of the larger rivers draining into the Gulf of Bothnia as well as offshore waters. A 3D hydrodynamic model makes it possible to take water movements into account, which cannot be neglected on longer time scales. A suite of modelling experiments is performed to describe the underlying processes behind the observed $pCO_2$. The objectives of this study are to investigate if, and in that case on what time scale, remineralization of tDOC into DIC is needed to explain the observed high $pCO_2$ values in the coastal waters, or if input of $CO_2$ supersaturated river water is enough to explain this pattern. Because there is no clear consensus on which is the dominating remineralization process in the Baltic Sea, it is parameterized as a simple linear decay (after Fransner et al. (2016)) that is assumed to include the effects of both bacterial and photochemical remineralization. We further investigate the potentially damping effect the tDOC can have on the primary production and the $pCO_2$ drawdown by increasing the light

attenuation in nearshore waters. We conclude by looking at the impact of the tDOC on the air-sea $CO_2$ exchange in the Gulf of Bothnia and weather it turns it to a net heterotrophic system.

## 2 Methods

### 2.1 Model setup

The model setup used for this study (BFM-NEMO-GoB) consists of a 3D coupled hydrodynamical-biogeochemical model applied to the Gulf of Bothnia (GoB, Figure 1), (Fransner et al., 2018). It has approximately two nautical miles (3704 m) horizontal resolution and 36 vertical levels with increased resolution towards the ocean surface. An open boundary towards the Baltic Proper is located in the Southern part of the domain at 59.9 °N (Figure 1). The hydrodynamical part is based on the NEMO-Nordic model (Hordoir et al., 2013, 2015, 2018), which is a NEMO 3.6 (http://www.nemo-ocean.eu, Madec and the

NEMO team (2016)) configuration for the Baltic and the North Seas with the LIM3 sea ice model (Vancoppenolle et al., 2009). The performance of the sea ice dynamics in NEMO-Nordic is validated in Pemberton et al. (2017). A comparison between modelled and observed sea ice concentration climatologies can also be found in Figure S1 in the supplementary material. BFM-NEMO-GoB is driven by hourly downscaled ERA40 data (Samuelsson et al., 2011), and river runoff from the EHYPE model (Donnelly et al., 2016). The biogeochemical part consists of the Biogeochemical Flux Model (BFM; http://bfm-community.eu)

(Vichi et al., 2007a, 2015a). BFM is a stoichiometric model that simulates the biogeochemical cycles of carbon (C), nitrogen (N), phosphorus (P) and silica (Si). It has four phytoplankton groups, four zooplankton groups (partitioned into micro and meso-zooplankton), one group of bacteria, particulate organic matter, and two groups of dissolved organic matter of different lability. A separate functional group representing terrestrial dissolved organic matter has been added to the BFM-NEMO-GoB setup (Fransner et al., 2018). While the organic matter that is a built in feature in BFM is degraded by bacteria, the terrestrial

dissolved organic matter is subject to a linear decay, which will be further described in section 2.3. The forcing data for the biogeochemical part consists of river inputs of inorganic and organic C,N,P, Si, and total alkalinity, as well as atmospheric depositions of DOC, phosphate and inorganic and organic nitrogen. The riverine loads have been calculated by multiplying measured concentrations of the chemical species with the volume flow in EHYPE (Fransner et al., 2016, 2018). The riverine input of organic carbon is supposed to consist of 10% particulate organic carbon (POC) and 90% DOC (Fransner et al.,

2016, 2018). As in Fransner et al. (2016), the DOC of atmospheric origin is considered as tDOC. A complete description and evaluation of the BFM-NEMO-GoB setup, including the mean seasonal $pCO_2$ cycle, can be found in Fransner et al. (2018).

### 2.2 pCO$_2$ data

The $pCO_2$ was measured during 25 cruises, spanning January to October 2012, with the TransPaper cargo (Fransson et al., in preparation). The TransPaper cargo sails from Gothenburg on the Swedish west coast, through the Baltic proper and northwards

through the Bothnian Sea and the Bothnian Bay to the ports of Oulu and Kemi in Finland. The $pCO_2$ data were gained by infrared analysis of equilibrator headspace samples. The specific instrument was supplied by General Oceanics®and designed

following the principles presented by Pierrot et al. (2009) using two-stage showerhead equilibration and a LICOR®7000 non-dispersive infrared detector. The system was calibrated using four high-qualitative reference gases with approximate values of 250, 350, 450 and 550 ppm, traceable to reference standards (National Oceanic and Atmospheric Administration – Earth System Research and Laboratory), see Pierrot et al. (2009) for a more detailed description of the system. The seawater was supplied from an intake located mid-ships, at approximately 7 m water depth. Temperature was recorded in the surface-water intake using a Seabird CTD and in the equilibrator using 1521 temperature probes from Hart Scientific, with an accuracy of 0.01 °C. The mole fraction of $CO_2$ ($xCO_2$) in the atmosphere was measured in air samples, pumped from an air intake located at approximately 50 m above sea level, where contaminated samples were removed. Air pressure was recorded by a high precision Druck barometer mounted at the air intake.

The measured $pCO_2$ and the cargo route for every month are displayed in Figure 2.

## 2.3 Simulations

The experiments have been performed in three sets (Table 1). In the first set, containing two experiments, all terrestrial organic carbon (both particulate and dissolved) is excluded. The first experiment (CHEM) investigates whether river water oversaturated in $CO_2$ can explain the high $pCO_2$ in the low-salinity region. It is done by excluding all biological processes in the water column and in the sediments. The model is thus only computing the carbonate chemistry. The only processes affecting the state of the carbonate chemistry in this experiment are river discharge of total alkalinity and DIC, air-sea exchange, and changes in temperature and salinity (due to riverine and atmospheric forcing). In the second experiment, BIO, the biological processes are activated, to see whether remineralization of autochthonous organic carbon, both in the sediments and in the water column, can explain the waters oversaturated in $CO_2$.

In the second set, the remineralization experiments (Table 1), the remineralization kinetics of riverine POC and DOC are examined by running three experiments, TP, 1Y and 10Y. The aim of these experiments are to investigate whether remineralization of tPOC is enough to explain the high $pCO_2$ in the low-salinity region, or if a remineralization of tDOC (and in that case on what times scale) is needed. The TP experiment is the same as the BIO experiment, with the addition of the supply of terrestrially derived POC. For simplicity we haven't added a separate group for terrestrial POC and it is therefore subject to the same dynamics as the autochthonous POC, meaning that it is degraded by bacteria with a time scale of 10 days. As the terrestrial POC only consists of 10% of the total load of riverine organic carbon, this assumption does not have significant impact on our results.

The 10Y and 1Y experiments are the same as the TP experiment, but with the addition of tDOC subject to a linear remineralization rate. These experiments are based on Fransner et al. (2016) who showed, by using passive tracer representing tDOC in a 3D physical model of the Baltic Sea, that observed concentrations of tDOC in the Baltic Sea (Deutsch et al., 2012) can be obtained with two different parameterizations of tDOC removal. In the first parameterization, a decay rate on of the time scale of ten years was applied to 100% of the tDOC entering the Baltic Sea. In the second one, 20% of the tDOC was assumed to be refractory (resistant to removal), and 80% was assumed to be labile subject to a decay rate on the time scale of 1 year. Here we apply the same experiments in a biogeochemical model. Because tDOC can be remineralized by both bacteria and

solar radiation, and there is no clear consensus on which of these are the dominating process in the Baltic Sea, we use the same linear decay as in Fransner et al. (2016) that is assumed to include the effect of both of these processes, instead of letting it be degraded by the bacteria in the model. In the 1Y experiment (similar to the REF experiment in Fransner et al. (2018)) a decay constant of 1 $y^{-1}$ is applied to 80% of the tDOC (the labile pool) entering the Gulf of Bothnia, and the remaining 20%

is assumed to be refractory. The refractory part of the tDOC is not modelled explicitly, and is removed from the river load. In the 10Y experiment a decay constant on the time scale of 10 years is applied to the whole pool of tDOC. The remineralized tDOC goes directly to the DIC pool. Terrestrially derived organic nutrients have been shown to be important nutrient sources for phytoplankton in the Baltic Sea (Stepanauskas et al., 2002). The input and degradation (with a degradation rate on the time scale of one year) of terrestrial organic nutrients (Fransner et al., 2018) are constant over all experiments and all three sets to

make sure that any differences in pCO$_2$ is not caused by changes in primary production.

     The third set contains one experiment (1YS) that is similar to the 1Y experiment, but where a tDOC dependent light parameterization is used instead of a salinity dependent one (equation A4 in Fransner et al. (2018)). The aim of 1YS is to investigate the potential indirect effect tDOC could have on the pCO$_2$ by dampening phytoplankton growth and carbon fixation. Unfortunately, there are little data available of simultaneously measured DOC concentration and photosynthetic available radiation.

We have therefore created a simple parameterization where we let the tDOC-induced light extinction coefficient ($k_{d_{tDOC}}$) vary as a linear function of the labile tDOC according to:

$$k_{d_{tDOC}} = 0.15 + 10^{-3} tDOC_l \qquad (1)$$

where tDOC$_l$ is the concentration of the labile tDOC in $mg$ C m$^{-3}$, the constant $10^{-3}$ has the units [m$^{-1}$ ($mg$ C)$^{-1}$ m$^3$], and $k_{d_{tDOC}}$ has the units [m$^{-1}$]. This means that $k_{d_{tDOC}}$ is 0.15 at zero labile tDOC concentration, and amounts to 7.5 close to

river mouths. The reason for $k_{d_{tDOC}}$ to be 0.15 at zero concentration of labile tDOC is to take into account the contribution of the refractory tDOC, which is not modelled explicitly in our experiments. $k_{d_{tDOC}}$ is together with the modelled chlorophyll-a and POC concentration modulating the total light extinction coefficient $k_d$, which ranges from 0.23 to 7.6 in surface waters (Figure 3). Ask et al. (2009) measured light extinction coefficients up to 4 in Swedish lakes, and Arst et al. (2008) measured as high as 10 at about the same maximum DOC concentrations as in Finnish rivers that drain into the Gulf of Bothnia, suggesting

that our modelled $k_d$ lies within a reasonable range. The tDOC dependent light parameterization results in a steeper gradient in the light extinction coefficient between coastal and offshore waters than in the 1Y experiment (Figure 3). While $k_d$ in the middle of the basins is rather similar in the two simulations, the $k_d$ is much larger in the coastal waters in the 1YS experiment.

     All simulations are run for 20 years, from 1990 to 2010, and the output data are saved at a monthly frequency. The simulations are started from restarts after a 10 year spinup (REF experiment in Fransner et al. (2018)). Climatological means (20 years) of

the simulations are compared to the observed pCO$_2$. The comparison between modelled and observed pCO$_2$ will be done in salinity space as the influence of river discharge on the pCO$_2$ becomes more apparent with these coordinates. Maps of modelled salinities are shown in Figure S2 in the supplementary material.

## 3 Results

### 3.1 Description of observed $pCO_2$

There is a strong seasonal as well as spatial variability in the observed $pCO_2$ (Figure 2). In January to March rather high $pCO_2$ values of 400-500 $\mu$atm are observed in the offshore areas. In the North-Eastern parts of the Bothnian Bay, supersaturated waters of up to 1500 $\mu$atm are observed. In April the spring bloom begins in the Bothnian Sea and patches of undersaturated waters can be observed. The waters in the Bothnian Bay stay oversaturated. In May, the waters are undersaturated in $pCO_2$ in the Bothnian Sea, and oversaturated in the Bothnian Bay. The waters in the North-Eastern parts of the Bothnian Bay stay highly oversaturated (>1000 $\mu$atm) in the observations also in April and May. During June and July the waters in almost the entire domain are undersaturated. The waters in the North-Eastern parts are however slightly oversaturated. In August the $pCO_2$ starts rising due to a combination of lower productivity and mixing/entrainment of $CO_2$ rich deep water, and in October it returns to to 400-500 $\mu$atm. In the North-Eastern Bothnian Bay no $CO_2$ supersaturated ($pCO_2$>1000) waters are found during September and October. During November and December no observational data exists.

The influence of river water on the $pCO_2$ becomes clearer in salinity coordinates (i.e. if plotting the $pCO_2$ against salinity instead of in lat-lon coordinates, Figure 4). A distinct decrease of $pCO_2$ with increasing salinity is observed especially from January to May. High $pCO_2$ values well above 1000 $\mu$atm are observed at salinities below 3. The $pCO_2$ values in this low-salinity region (0-3) are scattered, but there seems to be a general pattern with two branches, one with higher $pCO_2$ and one with lower. They might correspond to whether the ship was breaking through compact sea ice or going in an already open channel, respectively. Also in June and July there is a clear decrease of $pCO_2$ with salinity, although the $pCO_2$ in the low-salinity region is not as high as during the first five months of the year. In August the $pCO_2$ values in the low-salinity region are rather scattered. In September and October no $pCO_2$ measurements exist in the waters with the lowest salinity.

### 3.2 High $pCO_2$ river water and marine OC

When comparing modelled $pCO_2$ in the CHEM experiment with the observations it becomes clear that discharge of river water oversaturated in $CO_2$ cannot explain the observed high $pCO_2$ values in the low-salinity region (Figure 4). The influence of river water on $pCO_2$ is overall negligible for the $pCO_2$ dynamics in the Gulf of Bothnia, and the modelled $pCO_2$ in the CHEM experiment is close to atmospheric equilibrium, with the exception of temperature effects that causes a seasonal variation in the $pCO_2$ of up to 100 $\mu$atm.

When activating the biology and the autochtonous production of organic carbon (the BIO experiment), as well as the water-sediment interaction, the model simulates a slight oversaturation of $CO_2$ in the low-salinity region during January-April (Figure 4). It is, however, not high enough to explain the observed $pCO_2$ values. During summer the model draws down the $pCO_2$ too much in the low-salinity area, which could either be a result of too little remineralization, or a too high primary production.

### 3.3 Remineralization of terrestrial OC

When adding river discharge of highly degradable terrestrial POC (tPOC) to the BIO setup (TP experiment), the model simulates the lower branch of the observed $pCO_2$ in the low-salinity region from January to March (Figure 5). It is however not enough to explain the observed high $pCO_2$ values, indicating that there is not enough remineralization in this area.

Subjecting tDOC to a decay, as in the 1Y and 10Y experiments, results in higher remineralization per volume unit where the highest concentrations of tDOC occur. Consequently, in the North-Eastern Bothnian Bay, where the highest tDOC concentrations are found (not shown here, but in Fransner et al. (2016)), the remineralization rates are also the highest (Figure 6). It is in the areas with the highest remineralization that the largest impacts on the $pCO_2$ are seen (Figure 2 and 6). Adding remineralization of tDOC results in an increase in $pCO_2$ by up to 350 in the coastal waters in the 1Y experiment, while the

increase is only 80 in the 10Y experiment, on annual average (Figure 6).

As seen in Figure 5, the 1Y experiment reproduces the observed $CO_2$ supersaturated (>1000 $\mu$atm) waters in spring, although it does not capture the highest observed values. The 10Y experiment results in higher $pCO_2$ than the TP experiment in the low-salinity region, but the differences are small, and it barely simulates a $pCO_2$ above 1000 ppm, except at the lowest salinities. Interestingly, the high $pCO_2$ values above 1000 $\mu$atm only exist during periods when there is sea ice, both in the 1Y experiment

and in the observations. When removing the damping effect of sea ice on the air-sea $CO_2$ exchange, the 1Y experiment no longer simulates the higher $pCO_2$ values, and the simulated $pCO_2$ values in the low-salinity region approach the ones in the TP and 10Y experiments (Figure S3 in Supplementary Material). This is an additional indication that the two observed branches in the $pCO_2$ during the ice-covered months could be a result of whether the ship has travelled through open or ice-covered water.

During the productive season, none of the remineralization experiments, not even the one with a higher degradation of tDOC,

is capable of reproducing the higher $pCO_2$ values in the low-salinity region (Figure 5 e-h). This is probably due to a too high productivity, which will be discussed in Section 3.4.

### 3.4 Terrestrial DOC and light extinction

Adding a linear dependency of the light extinction coefficient on the tDOC concentration, as in experiment 1YS, gives a steeper gradient in the light availability between coastal and offshore waters (Figure 3). The reduced light availability decreases the

primary production and nutrient consumption in coastal areas (Figure 7), which results in a larger transport of nutrients offshore, partly explaining the increased primary production in the middle of the basins. The parameterization of the light extinction coefficient in the 1YS also results in slightly clearer waters in the middle of the basins, which also increases the primary production. The tDOC dependent light extinction has the largest effect in the Bothnian Bay, where the primary production is reduced by 25% (Table 3). In the Northern Quark and the Bothnian Sea, as well for the whole domain, there is barely any

change in the total primary production.

The lower primary production in the coastal areas in the 1YS experiment leads to elevated $pCO_2$ in these areas (Figure 7). In the low-salinity region, the $pCO_2$ stays oversaturated also during the summer period (Figure 8), and agrees better with observed $pCO_2$ than the 1Y experiment does. During the winter months the $pCO_2$ in the low-salinity region is slightly decreased. The

decrease is caused by the lower primary productivity and consequently the reduced export of organic carbon to the sediments, which leads to a lower DIC (Dissolved Inorganic Carbon) efflux from the sediments. A comparison of the simulated $pCO_2$ in 1YS with observations in geographical space is shown in Figure 9. It shows an overall good agreement with the observations. The largest discrepancies are found in the Bothnian Sea in March and September and are related to the onset of the spring bloom and the autumn mixing, respectively, which causes relatively large changes in $pCO_2$ over a short period of time. Both of these discrepancies can be related to that the model results show a monthly mean, while the measurements have been taken in the first half of the month for March, and second half of the month of September, respectively. The measurements are therefore biased towards the period of high $pCO_2$ in both March and September. The tDOC dependent $k_d$ parameterization also results in a better agreement between modelled an observed seasonal cycles of nutrients in the North-Eastern Bothnian Bay (Figure S4 and S5 in supplementary material). In the middle of the basins (the stations in Figure 1) the difference between the 1Y and 1YS experiments are small (Figures S6-S19 in the supplementary material).

## 4 Discussion

### 4.1 Remineralization of terrestrial DOC

Our results clearly show that input of river water over-saturated in $CO_2$ is not enough to explain the high $pCO_2$ values observed in the Northern Gulf of Bothnia, and suggest that it is a result of a substantial remineralization of tDOC into DIC. Here we tried two different rates of remineralization, one on the order of 1 year applied to 80% of the tDOC, and one on the order of 10 years applied to 100% of the tDOC. These removal rates were derived in a 3D model (Fransner et al., 2016) to simulate observed concentrations of tDOC in the Baltic Sea (Deutsch et al., 2012). We showed here that only the simulation with the faster rate was able to reproduce the $CO_2$ supersaturated waters, although it didn't capture the highest observed values. The reason for this could be that there are more labile pools (with faster degradation rates) of the tDOC that we do not resolve in our relatively simple model. It is well known that organic matter consists of a continuum of pools with different lability that are subject to different remineralization rates (Hansell, 2013; Carlson et al., 2015). Pools with faster remineralization rates than the one we use would be remineralized closer to the river mouth, and therefore cause higher $pCO_2$ at lower salinities.

Considering that the removal rate of tDOC in the 1Y experiment not only results in a good agreement between observed and modelled concentrations of tDOC, as shown in Fransner et al. (2016), but also results in a good agreement with observed $pCO_2$ values, it suggests that remineralization of tDOC into DIC is the main mechanism behind tDOC removal in the Gulf of Bothnia. In other words, flocculation into particulate organic carbon seems only to play a minor role in removal of tDOC from the water column, which also was suggested by Asmala et al. (2014b). The high $pCO_2$ values observed during the ice season, when there is little light reaching the surface water, would further argue against photochemical degradation as the main removal mechanism, in contrast to what was suggested by Aarnos et al. (2012). Incubation experiments do however suggest that only 10-20% of the terrestrial DOC is available to bacterial degradation (Wikner et al., 1999; Asmala et al., 2013, 2014a; Herlemann et al., 2014; Hulatt et al., 2014; Figueroa et al., 2016). The time scale of these incubation experiments are on the

other hand relatively short (on the order of weeks to a few months), and the availability could be larger if exposing the DOC to bacteria during a longer period of time, as discussed in Fransner et al. (2016).

Knowing the incubation length in time, and the relative change in DOC concentration, a average degradation rate of tDOC during the time of incubation can be calculated based on the the classical expression for exponential decay:

$$C = C_0 e^{-\lambda t} \tag{2}$$

where $\lambda$ is the decay constant (degradation rate), $t$ is the incubation length in years, $C$ is the concentration of DOC at the end of the incubation and $C_0$ is the concentration of DOC at the start of the incubation. Rearranging Equation 2, an expression for $\lambda$ is obtained:

$$\lambda = -\frac{1}{t} log\left(\frac{C}{C_0}\right) \tag{3}$$

Interestingly, when calculating the degradation rates for various published incubation experiments from the Gulf of Bothnia, many of them are on the time scale of the order of one year (Table 2), the same time scale that we use for the degradation in our 1Y experiment[1] This indicates that bacteria could be capable of remineralizing similar amounts of tDOC as in our experiments and in Fransner et al. (2016) (80% of the load to the Baltic Sea), if only considering longer timescales than those of the incubation experiments. This is in line with what was suggested by Kuparinen et al. (1996) and Sandberg et al. (2004) who, based on extrapolations of the carbon demand of secondary producers, suggested that a large part of the tDOC entering the Gulf of Bothnia is degraded by bacteria. Table 2 gives furthermore an additional indication that the 1Y experiment is more realistic than the 10Y experiment.

Equation 3 is a very simple model of degradation; organic matter tends to consist of several pools subject to different degradation rates (Hansell, 2013; Carlson et al., 2015). Three of the incubation experiments in our comparison (Table 2) have several sampling points in time that indicate that the degradation rate decreases with time, and that the tDOC consists of more than two pools of different lability in contrast to our experiment 1Y (Asmala et al., 2014a; Herlemann et al., 2014; Hulatt et al., 2014). Hulatt et al. (2014) for example, calculate the degradation rates with a continuum model at different times during the incubation and report degradation rates on the order of 3 months in the beginning and 5 years in the end of their experiments (after 55 days). When working on larger spatial scales such as our model and our *in situ* measurements cover, it is however difficult to go into these fine details of degradation dynamics.

There are further many aspects that complicate a detailed comparison between results from incubation experiments to what is happening in the natural environment (from which we use observations to compare our model results to). Incubations are rather artificial environments, where effects of turbulence, stratification, sunlight and interactions with other organisms/chemical

---

[1]In contrast to the model of remineralization in 1Y that have two pools of tDOC of different lability, Equation 3 only considers one pool. The results of these two models are comparable until the labile pool starts to be depleted (after one year), which is why we can compare the degradation rate of the labile pool in 1Y and the degradation rates calculated for incubation experiments with a time duration up to 55 days.

constituents often are absent (depending on the experimental setup). It has been suggested, for example, that the lability of relatively refractory organic matter can increase in presence of more labile substrates (priming) (Bianchi, 2011; Blair and Aller, 2011) and solar radiation (Vähätalo et al., 2011), which would not occur in incubation experiments.

## 4.2 Terrestrial DOC and light extinction

The results from the 1YS experiment show that a strong extinction of light induced by terrestrially derived organic matter, hampering the primary production, could explain why the waters stay oversaturated in $pCO_2$ in summer. It doesn't only improve the modelled $pCO_2$, but also results in a better agreement between modelled an observed seasonal cycles of nutrients in the North-Eastern Bothnian Bay (Figure S4 and S5 in supplementary material), further suggesting that this parameterization is reasonable. The measurements in Arst et al. (2008) and Ask et al. (2009) show a large spread in the light extinction coefficients

for different lakes, and we based our parameterization on the upper values that they present. We also made experiments where the effect of tDOC on the light attenuation was weaker, and where $k_d$ reached up to one at the lowest salinities. This was however not strong enough to prevent a too large drawdown of the $pCO_2$. Local measurements of $k_d$ and DOC would be needed to better understand the influence of tDOC on light attenuation in the Gulf of Bothnia, and to create a more precise parameterization.

Although the tDOC-dependent light parameterization has an overall negligible effect on the primary production in the Gulf of Bothnia (Table 3), it has quite large local effects. The primary production is reduced in coastal waters, leading to a larger offshore transport of nutrients. The filtering effect of coastal waters (Asmala et al., 2017) is thus decreased. Clearly, more measurements of the relationship between light and DOC are needed to better understand not only the carbon fixation in coastal waters, but also the exchange of nutrients between coastal and offshore waters.

## 4.3 The influence of terrestrial DOC on the air-sea $CO_2$ exchange

The remineralization of tDOC in the 1Y experiment reduces uptake of atmospheric $CO_2$ by in total 43% (Table 4), compared to the simulation with no terrestrial DOC (TP-simulation). The reduction in the atmospheric $CO_2$ uptake (17.5, 8.3, 6.7, 10.0 $m^{-2}$ $y^{-1}$) corresponds well to the amount of remineralized tDOC in each subbasin (18.2, 8.2, 6.6 and 10.1 mg $m^{-2}$ $y^{-1}$ for BB, NQ, BS and the whole domain, respectively), indicating that almost all of the remineralized tDOC is outgassed to the

atmosphere, and that a negligible fraction of the remineralized DOC (1%) adds to the DIC pool. A surplus of remineralized DIC is transported from the BB to the southern basins, which is why there is a slightly larger reduction in atmospheric uptake in these basins than calculated from the remineralized tDOC. The large amount of remineralized tDOC in the Bothnian Bay turns it to a source of atmospheric $CO_2$ (Figure 10), in agreement with estimations by Löffler et al. (2012). However, the modelled outflux of $CO_2$ to the atmosphere in the Bothnian Bay is larger than their estimations. The simulated air-sea exchange in the

1Y and 1YS experiment agrees overall better with the estimations by Löffler et al. (2012), than the simulation without any remineralization of tDOC, strengthening our findings that a remineralization of tDOC into DIC takes place.

Adding a dependency of the light extinction on the tDOC increases the heterotrophy of the nearshore areas and the Bothnian Bay. Compared to the 1Y experiment (Table 4 and Figure 10), the outgassing of $CO_2$ is increased by 28% in the Bothnian

Bay. In the central parts of the Bothnian Bay and the Bothnian Sea, on the other hand, the outgassing/uptake slightly decreases/increases due to the increased primary production in these areas. The overall effect on air-sea $CO_2$ exchange is minor with only a decrease of 4%.

## 4.4 Future studies

In this study we have shown that remineralization is an important pathway for terrestrial DOC entering the Gulf of Bothnia. Considering that there is a large remineralization taking place under the sea ice (arguing against photochemical degradation), and that the rate we find is comparable to degradation rates calculated from bacterial incubation studies, we argue that bacteria has the potential to be responsible for this large removal. This needs to be investigated further, and an interesting next step from a modelling point of view would be to let the bacteria degrade the tDOC within the model, and compare modelled bacterial

biomass/growth rates to measured ones. Interesting studies, that would be possible to perform with a stoichiometric flexible model, could for example be done on the quality of terrestrial DOM (based on its nutrient content), and on the competition for inorganic nutrients between bacteria and phytoplankton, which has been shown to be dependent on the availability of organic carbon relative to nutrients (e.g. Bratbak and Thingstad (1985); Joint et al. (2002); Thingstad et al. (2008)).

## 4.5 Uncertainty analysis

In shallow areas such as the North-Eastern parts of the Bothnian Bay, sediment fluxes have a particularly large impact on the carbon cycling and the air-sea $CO_2$. The highest sediment-water DIC flux in the model is found next to the river mouths. The maximum modelled sediment-water fluxes in the Bothnian Bay during winter, when DIC is accumulated under the sea ice, is about $200\,\mathrm{mg\,m^{-2}d^{-1}}$ in the 1Y experiment, which is in good agreement with Silvennoinen et al. (2008), who measured fluxes around 180-240 $\mathrm{mgC\,m^{-2}\,d^{-1}}$ in the mouth of river Temmesjoki at low temperatures (5 deg. C). The modelled sediment-DIC

fluxes in the more central parts of the basins further agree well with Winogradow and Pempkowiak (2014). They calculated a mean flux of 9.9 $\mathrm{mgC\,m^{-2}\,d^{-1}}$ from four stations in the Gulf of Bothnia. The mean flux in the model, calculated from the same four positions, equals 8.6 $\mathrm{mgC\,m^{-2}\,d^{-1}}$. A sensitivity experiment was performed to investigate the sensitivity of the results to sediment fluxes. It was similar to the TP experiment, but the permanent burial of carbon was turned off, which leads to a higher carbon content in the sediments, an consequently a higher remineralization and DIC efflux. This experiment almost

reproduced as high $pCO_2$ values as the 1Y experiment. However, the DIC efflux from the sediments was also much higher than observations; the maximum modelled sediment-water fluxes in the Bothnian Bay during winter amounted to $400\,\mathrm{mg\,m^{-2}d^{-1}}$, and the modelled DIC flux at the four stations in the more central parts of the basins amounted to $17\,\mathrm{m^{-2}\,d^{-1}}$, which is about double the flux in the 1Y experiment and the observations.

## 5 Conclusions

In this study the remineralization of terrestrial DOC, and its influence on the $pCO_2$ and the air-sea $CO_2$ exchange, is studied in the Gulf of Bothnia. It is done by combining results from a coupled physical-biogeochemical model together with high resolution underway measurements of $pCO_2$ data. Our conclusions are the following:

1. High $pCO_2$ values are explained by remineralization of terrestrial DOC, with a remineralization time scale of 1 year.

2. The remineralization rate agrees well with bacterial uptake rates of terrestrial DOC calculated from incubation experiments from the Northern Baltic Sea.

3. In addition to the terrestrial DOC remineralization, a high light attenuation induced by terrestrial DOC is needed to dampen the primary production and to reproduce the summer $pCO_2$.

*Code and data availability.* The BFM and NEMO source codes can be obtained at http://bfm-community.eu and http://www.nemo-ocean.eu, respectively. The input files needed to reproduce the simulations can be obtained upon request to the corresponding author (filippa.fransner@hotmail.se). The $pCO_2$ is a part of a bigger $pCO_2$ dataset of the Baltic Sea which will be presented (and made publicly available) in an article that is in preparation (Fransson et al., in preparation). Until then the data can be obtained upon request to Agneta Fransson (Agneta.Fransson@npolar.no). The nutrient data used to produce Figure S4 in the supplementary material comes from the ICES data portal
(http://ocean.ices.dk/Helcom/Helcom.aspx?Mode=1).

*Competing interests.* The authors declare that they have no conflict of interest.

*Acknowledgements.* We acknowledge the use of the BFM (http://bfm-community.eu) and NEMO (http://www.nemo-ocean.eu) models. The simulations were performed on resources provided by the Swedish National Infrastructure for Computing (SNIC) at the Triolith system. Per Pemberton at SMHI is acknowledged for sharing the observations of sea ice in the Baltic Sea. This work was partly funded by Baltic
Ecosystem Adaptive Management (BEAM), a strategic research program at Stockholm University, Sweden, and by the Swedish Agency of Environment (Naturvårdsverket). The Baltic Nest Institute is supported by the Swedish Agency for Marine and Water Management through their grant 1:11 - Measures for marine and water environment. L.T. acknowledges support from the BONUS COCOA project (grant agreement 2112932-1), funded jointly by the European Union and the Academy of Finland. We wan't to thank the two anonymous reviewers for their suggestions that significantly improved our work.

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

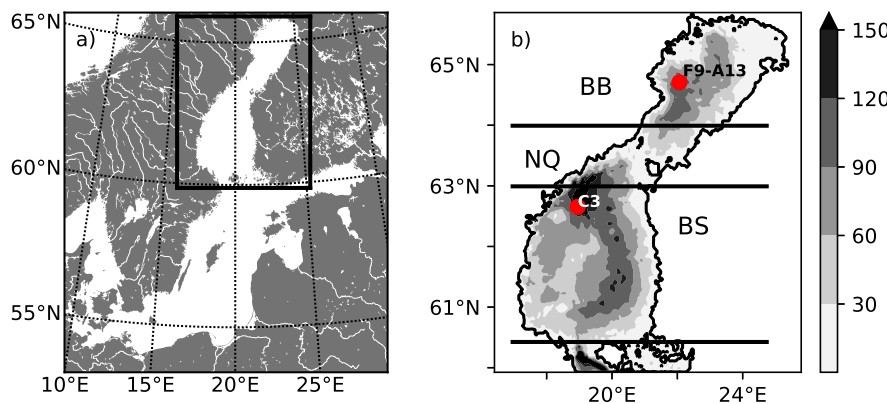

**Figure 1.** a) Map of the Baltic Sea. The rectangle marks the location of the Gulf of Bothnia and the model domain b) Bathymetric chart of the NEMO-GoB configuration. The filled contours show the depth (m). The horizontal lines marks the borders of the subbasins: the Bothnian Bay (BB), the Northern Quark (NQ) and the Bothnian Sea (BS). The two red dots show the position of two stations that are used for evaluation of the model (Figures S6-S19 in the supplementary material).

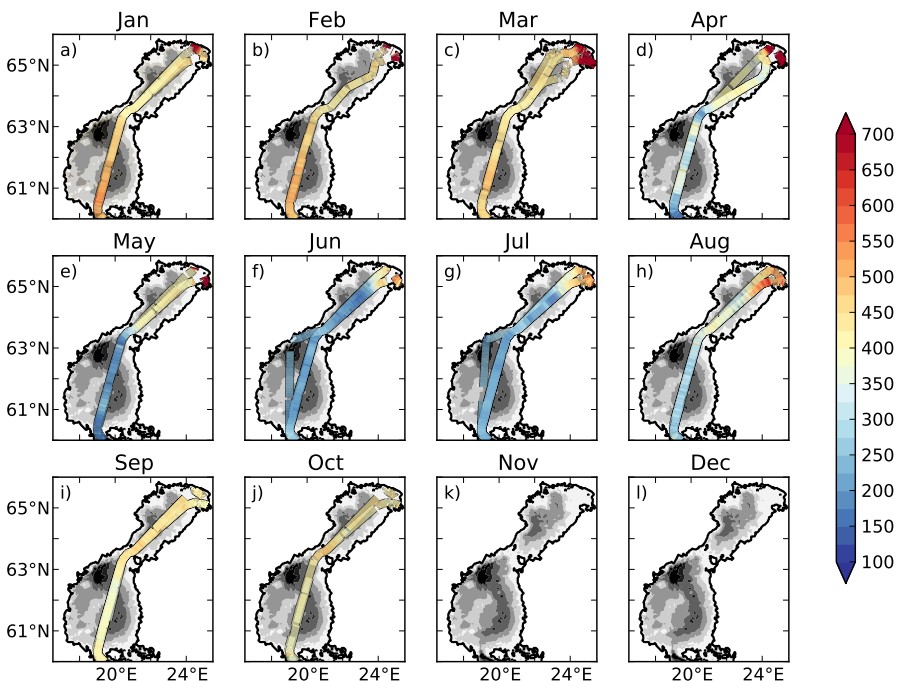

**Figure 2.** Observed (filled lines) pCO$_2$ ($\mu$atm) and cargo route for each month. The filled contours show the bathymetry of the model.

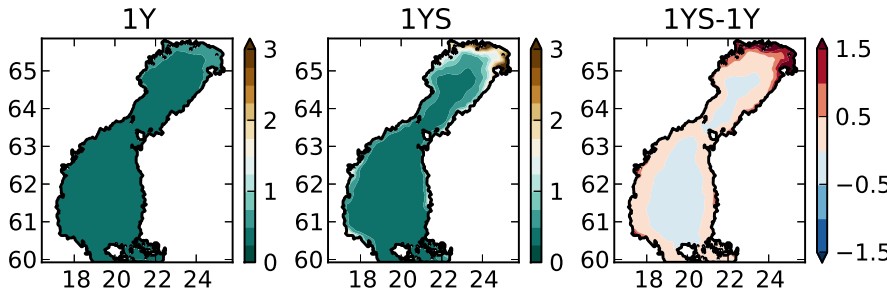

**Figure 3.** Modelled light extinction coefficient $(m^{-1})$ in the a) 1Y and the b) 1YS experiments, and c) the difference (1YS-1Y).

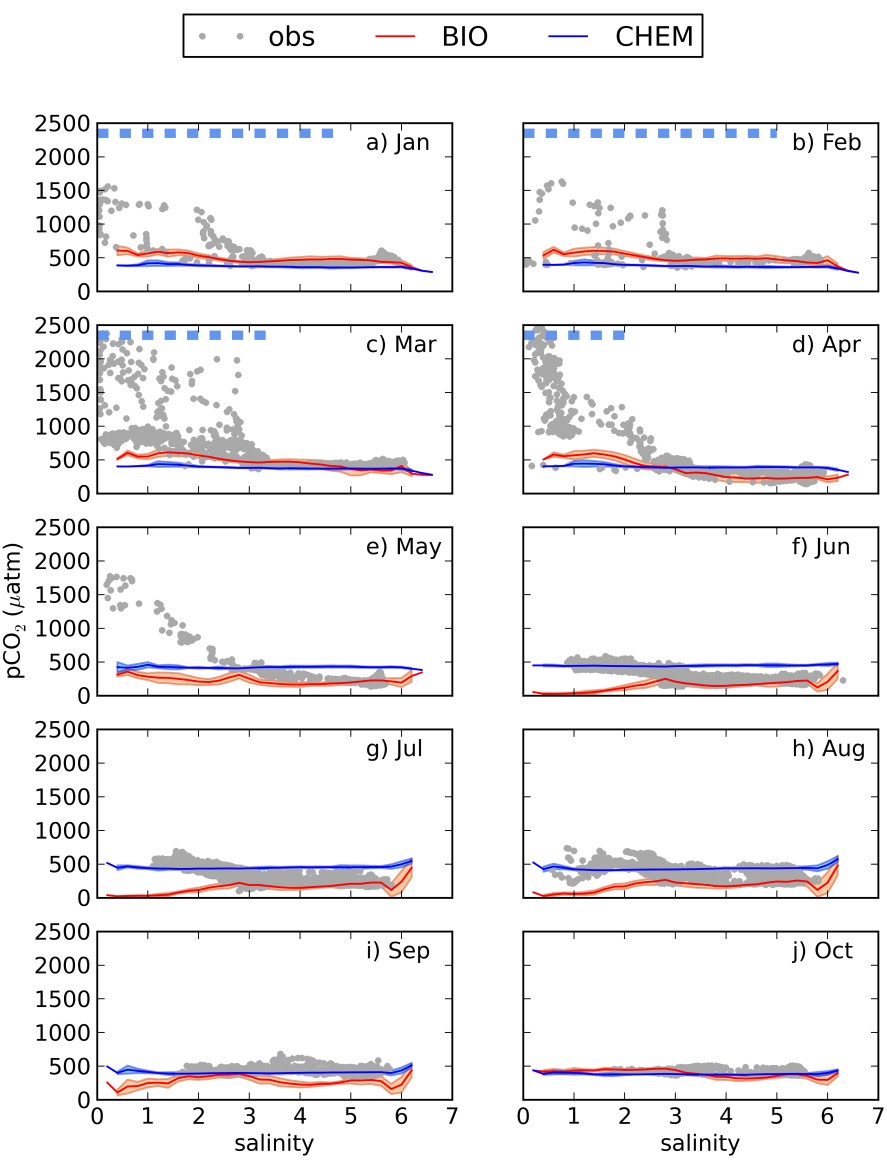

**Figure 4.** $pCO_2$-salinity relationships for January-October (a-j). Grey dots show observed values. The red and blue lines show modelled climatological monthly means for the BIO and the CHEM experiments, with the shaded area displaying the standard deviation at a given salinity. The dashed blue line shows the ice extent (salinities where the ice concentration is larger than 60%).

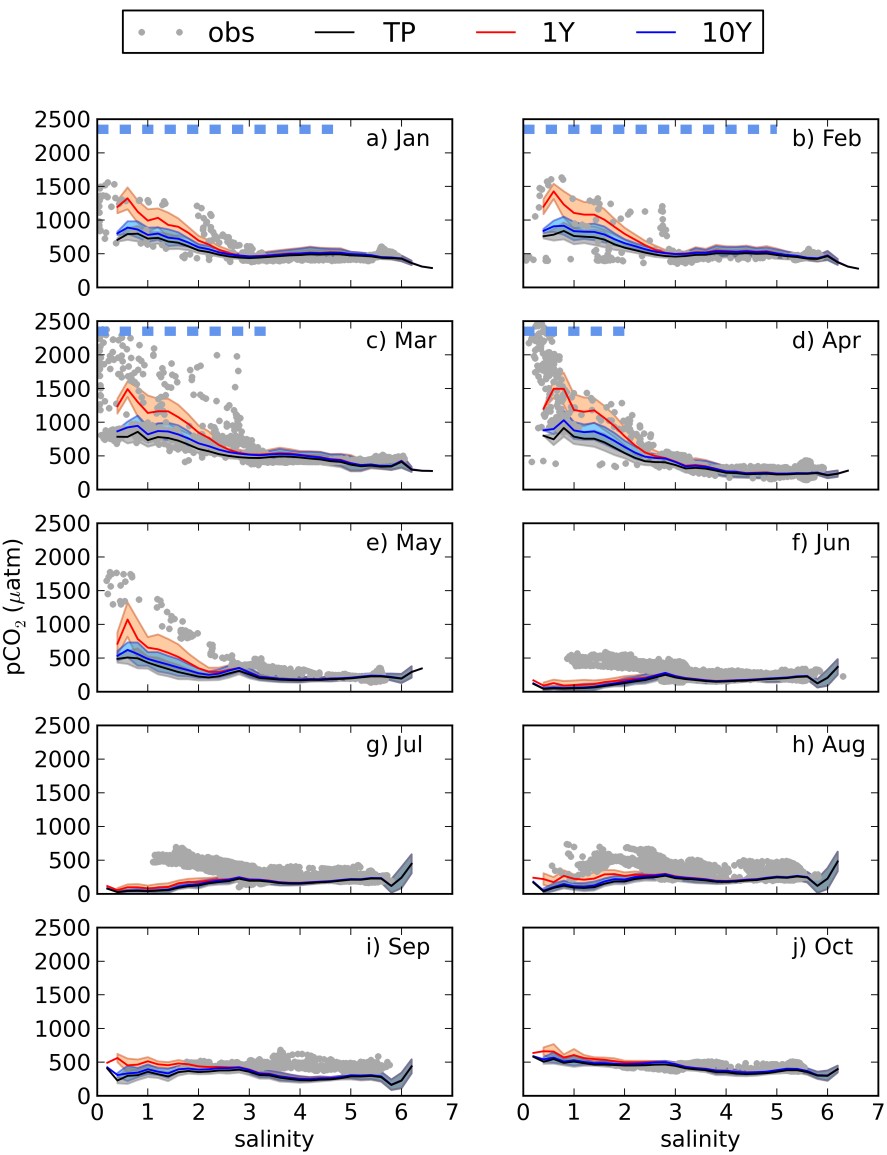

**Figure 5.** pCO$_2$-salinity relationships for January-October (a-j). Grey dots show observed values. The black, red and blue lines show modelled climatological monthly means for the TP, 1Y and 10Y experiments, with the shaded area displaying the standard deviation at a given salinity. The dashed blue line shows the ice extent (salinities where the ice concentration is larger than 60%).

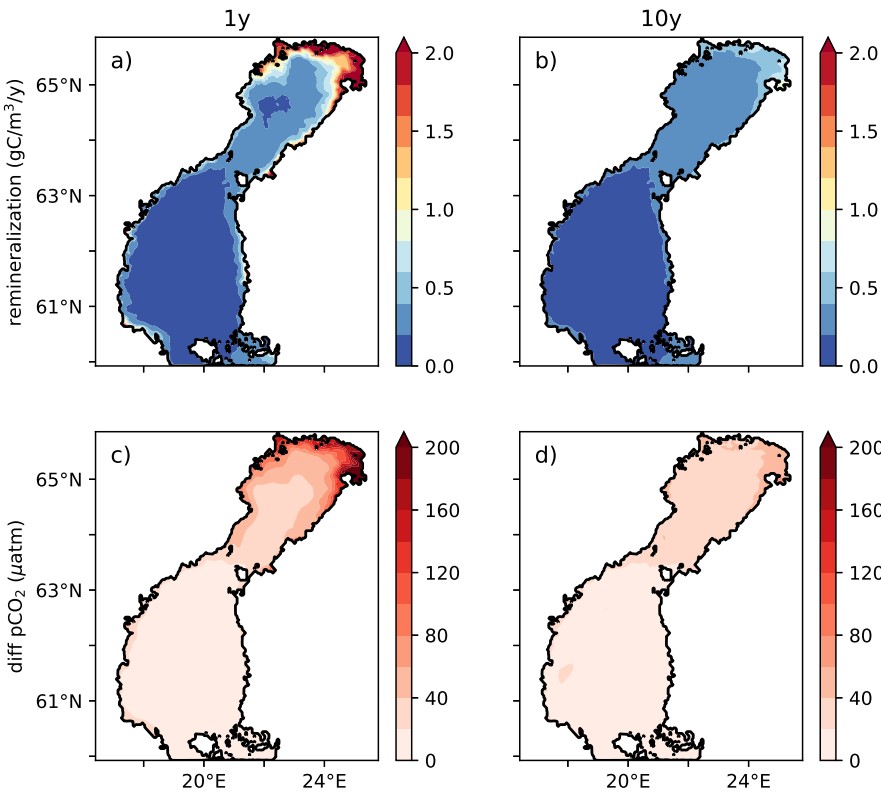

**Figure 6.** a),b) Vertically averaged remineralization rates of tDOC (g m$^{-3}$ y$^{-1}$) in the 1Y and 10Y experiment, respectively. c),d) difference in modelled pCO$_2$ ($\mu$atm), climatological annual mean, between the 1Y and the TP experiment, and d) the 10Y and the TP experiment, respectively.

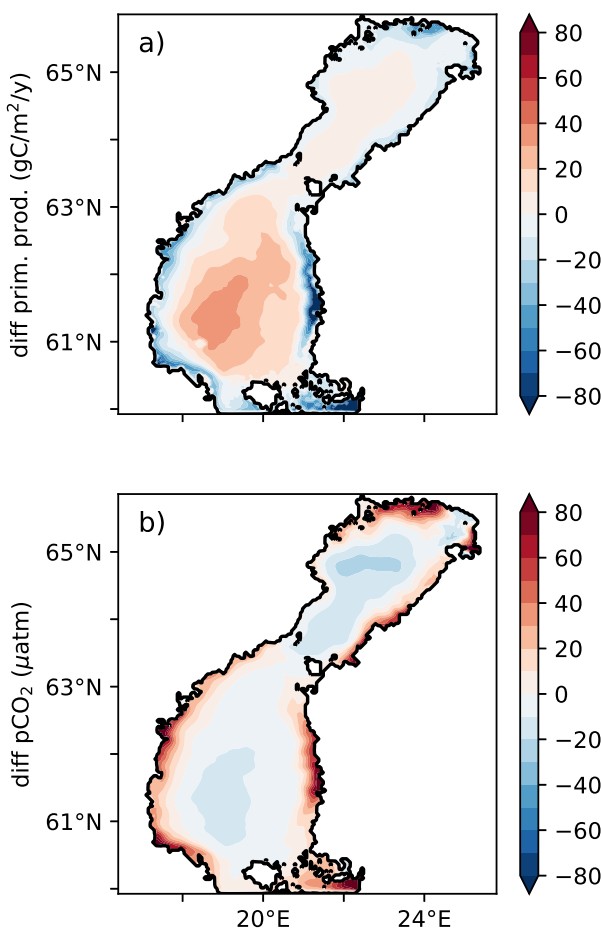

**Figure 7.** Difference in a) vertically integrated primary production (g m$^{-2}$ y$^{-1}$), and b) pCO$_2$ ($\mu$atm), between the 1Y and 1YS experiment.

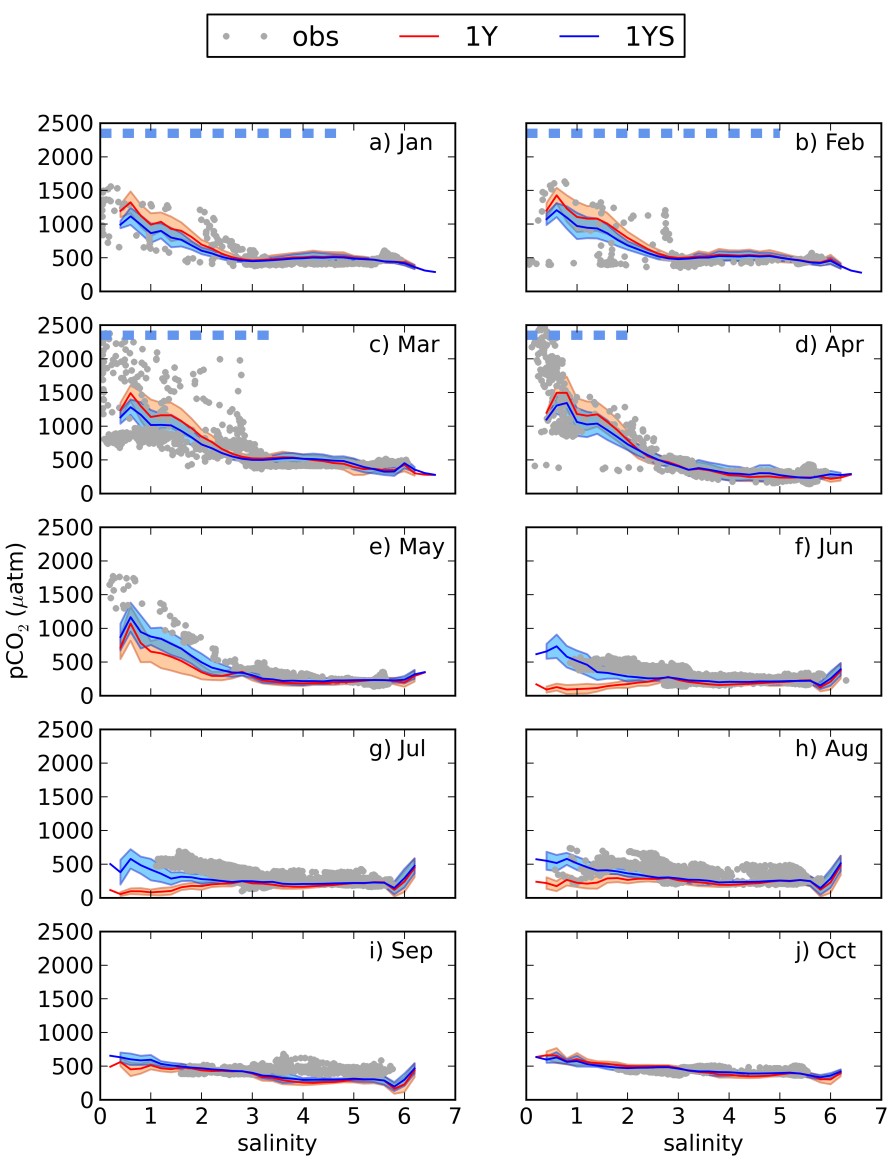

**Figure 8.** pCO$_2$-salinity relationships for January-October (a-j). Grey dots show observed values. The red and blue lines show modelled climatological monthly means for the 1Y and 1YS experiments, with the shaded area displaying the standard deviation at a given salinity. The dashed blue line shows the ice extent (salinities where the ice concentration is larger than 60%).

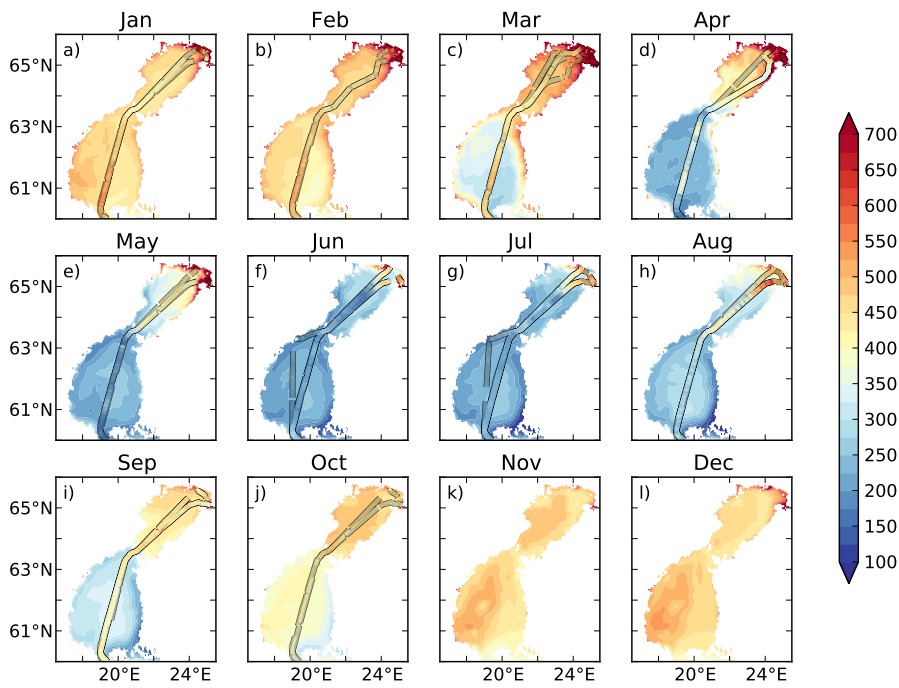

**Figure 9.** Observed (filled lines) and modelled (filled contours) pCO$_2$ ($\mu$atm) from the 1YS experiment.

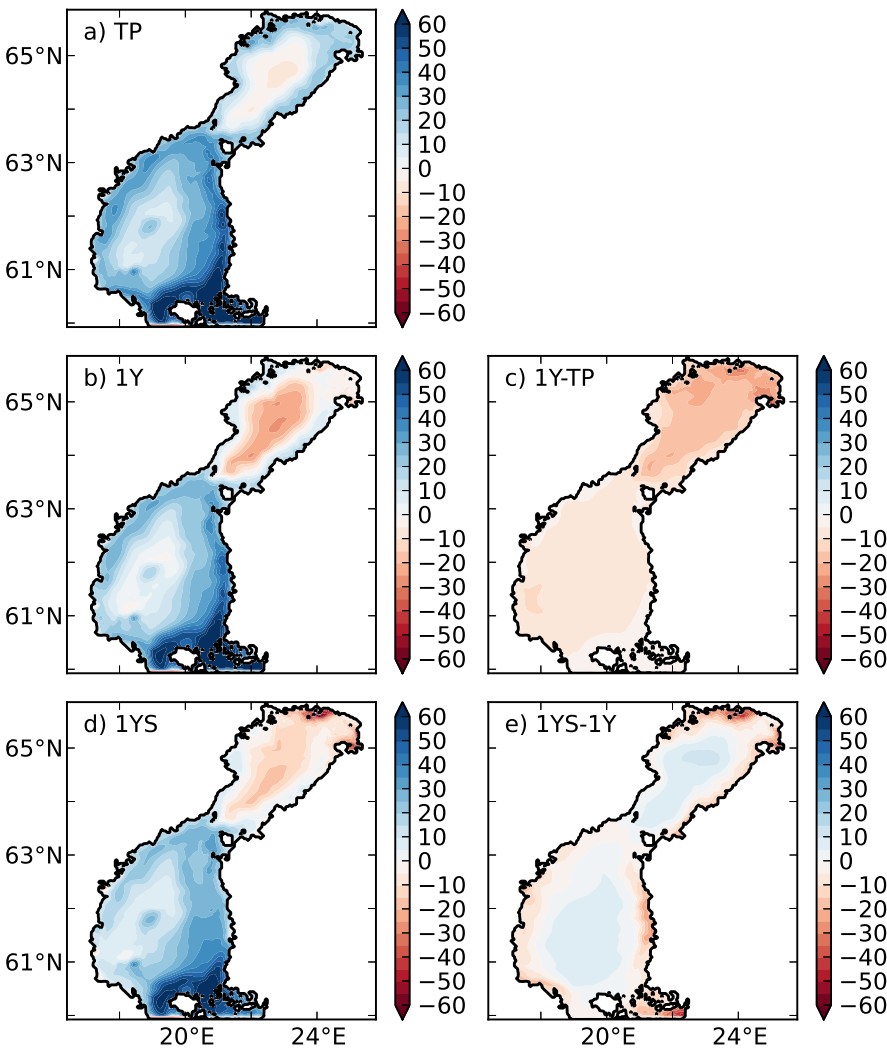

**Figure 10.** Air-sea $CO_2$ exchange (g m$^{-2}$ y$^{-1}$) in the a) TP, b) 1Y and d) 1YS experiments. Red indicates outgassing of $CO_2$ to the atmosphere, and blue uptake from the atmosphere. c) and d) show the difference in air-sea $CO_2$ exchange between the 1Y and TP experiments, and the 1YS and the 1Y experiments, respectively.

**Table 1.** Experimental setup

| Experiment | Activated modules | $\lambda^{-1}$ | $k_{tDOC}$ |
|---|:---:|:---:|:---:|
| **1st set.** | | | |
| CHEM | chem. | - | $f$(sal) |
| BIO | chem. & bio. | - | $f$(sal) |
| **2nd set (Rem exp.)** | | | |
| TP | chem., bio. & tPOC | - | $f$(sal) |
| 1Y | chem., bio., tPOC & tDOC | 1 year | $f$(sal) |
| 10Y | chem., bio., tPOC & tDOC | 10 years | $f$(sal) |
| **3rd set (Light exp.)** | | | |
| 1YS | chem., bio., tPOC & tDOC | 1 year | $f$(tDOC) |

The second column shows the activated modules in the biogeochemical model, where chem=
chemistry, bio= biology, and tPOC, and tDOC means that there is a remineralization of terrestrial
POC and DOC, respectively. The third column shows the remineralization time scale ($\lambda^{-1}$) of the
terrestrial DOC and the last column, $k_{tDOC}$, indicates whether the influence of the tDOC on the
light attenuation is a linear function of salinity (equation A4 in Fransner et al. (2018)) or tDOC
(equation 1).

**Table 2.** Removal of terrestrial DOC in incubation studies from the Gulf of Bothnia area.

| Sampling site | $t$ | % removed | $\lambda^{-1}$ | Reference |
|---|---|---|---|---|
| BB | 28 | 4–16 | 0.44–1.87 | Herlemann et al. (2014) |
| NQ | 6–15 | 6.3–8 (median) | 0.2–0.63 | Wikner et al. (1999) |
| GoB | 12–18 | 8.88 (mean) | 0.35–0.53 | Asmala et al. (2013) |
| BB | 39 | 9.0–13.5 (avg) | 0.7–1.3 | Asmala et al. (2014a) |
| BB | 10 | 2 (avg) | 1.35 | Figueroa et al. (2016) |
| BB | 55 | 9.8 (avg) | 1.46 | Hulatt et al. (2014) |

The first column shows the site of the sampling, where BB= Bothnian Bay, NQ= Northern Quark, and GoB is the whole Gulf of Bothina (Figure 1). The second column shows the length of the incubation in days and the third column shows the percentage of tDOC that has been removed at the end of the incubation (if average values are available these values has been reported, otherwise ranges). The fourth column shows the calculated time scale of degradation based on Equation 3.

**Table 3.** Primary production (1990–2010) in g C m$^{-2}$ y$^{-1}$ in the 1Y and 1YS experiments (relative change with respect to 1Y).

| Basin | BB | NQ | BS | GoB |
| --- | --- | --- | --- | --- |
| 1Y | 90 | 152 | 236 | 180 |
| 1YS | 71 (-25%) | 147 (-3%) | 240 (+2%) | 177 (-2%) |

**Table 4.** Air-sea $CO_2$ exchange (1990–2010) in g C m$^{-2}$ y$^{-1}$ in the TP, 1Y(relative change with respect to TP) and 1YS (relative change with respect to 1Y) experiments. Negative values indicates outgassing of $CO_2$ to the atmosphere, positive uptake of $CO_2$ from the atmosphere.

| Basin | BB | NQ | BS | GoB |
|---|---|---|---|---|
| TP | 10.9 | 24.5 | 29.4 | 23.3 |
| 1Y | –6.5 (-160%) | 16.2 (-34%) | 22.7(-23%) | 13.3 (-43%) |
| 1YS | –8.4 (-28%) | 15.7 (-3%) | 22.9 (+1%) | 12.9 (-4%) |
| Löffler et al. (2012) | –1.4- –2.5 | - | 17.05 | - |