# Peer review of "Remineralization rate of terrestrial DOC as inferred from CO2 supersaturated coastal waters"

_Biogeosciences, 2018_

## Referee Comment (RC1) · Anonymous Referee #1 · 13 Sep 2018

The manuscript describes a modeling exercise meant to investigate the fate of terrigenous DOC (tDOC) in the Northern Baltic Sea. More specifically, the authors provide an explanation to the high pCO2 observed in the area, concluding that this is due to the remineralization of tDOC by bacteria and the concomitant reduction in productivity due to the absorption of light by tDOC. The topic investigated is very interesting and the findings are relatively novel as the main removal process of tDOC was often assumed to be photo-degradation. The fate of the large amount of tDOC discharged into the ocean is still an open question in coastal oceanography and the results of this paper provide novel insights. The manuscript is clear and well written, the succession and explanation of the experiments are clear and logic. I would recommend publication

after the authors have considered/discussed/clarified the following points:

1) The authors use a quite complex biogeochemical model (BFM) which describes the planktonic ecosystem through a numbers of different plankton functional types. The latter include explicit bacteria and two species of DOM (labile and semi-labile). However, when considering tDOC, the authors use a simplistic decay function assuming that tDOC is all consumed in 1 or 10 years. Why tDOC was not assumed to be cycled by bacteria which are already modeled within the BFM? By using a fixed decay constant, remineralised tDOC goes directly into the DIC pool which is a simplification. Indeed the bacterial growth efficiency in estuaries and coastal zone is relatively high (del Giorgio and Cole 1998) implying that a substantial fraction of DOC assimilated by bacteria is incorporated into bacterial biomass. This might affect the ecosystem in various ways e.g. by affecting grazing (HNAN), the competition between bacteria and phytoplankton for nutrients and the production of recalcitrant DOC. It is also strange that the authors seem to use a different approach for the riverine POC which is assumed to be used by bacteria in the same way as marine POC. I think that the different approach-i.e. the lack of explicit bacterial utilization- used for the experiment with tDOC (the one leading to the main result of the paper) needs to be discussed and justified. 2) Equation 1(1). This equation is not very clear to me: If Kdtdoc represents the contribution of tDOC to the total light extinction, it should have the same units as the total Kd (i.e. m-1, as presented in Fig 3). The units reported at line 30 of page 4 seem to refer to the specific adsorption coefficient (see equation 9 in Vichi et al 2007) which (I guess) is represented by the parameter '1.0' multiplied by tDOC in eq 1. 3) tDOC is given in ïA■ug m-3 which is quite unusual for marine DOC (usually given in mmol m-3) this of course is not a big problem but from eq 1, tDOC concentrations seem to be very low (assuming a max value of Kdtdoc of 7.5). What is the concentration of tDOC given as input to the model? And what is the concentration of the simulated total DOC? 4) Why Kdtdoc is not equal to 0 when tDOC is zero? 5) The authors cited different papers reporting different light extinction coefficients (differing by more than one order of magnitude). This suggests that the parameters used to simulate kd are very uncertain. I think that a

sensitivity analyses would be useful to understand how the presented results (relative to the exp. 1YS) are affected by the choice of the specific light absorption parameters. 6) Only the labile fraction of tDOC is assumed to contribute to light extinction. However the biologically refractory fraction of tDOC can be composed by aromatic compounds which strongly interact with light (e.g Stubbins et al. 2010, L&O ) 7) No mention of the model skills in reproducing broad ecosystem variables (apart from DIN, DIP and pCO2) and fluxes. For example, is the primary production simulated in the various experiments realistic? Are there data available for comparison? If not, simulated values of Chl and primary production could be at least discussed in the context of what is observed in similar areas/ecosystems. I appreciate that the authors refer the reader to a previous paper for a complete validation of the model, however, it would be nice to see a summary of that validation in this manuscript. Additionally it would be very useful to see how the model performance varies in the different experiments reported here. For example, is chl and primary production simulated in exp 1yS more realistic than in the other scenarios investigated?. Without such (at least qualitatively) comparison the reader remains uncertain about the robustness of the conclusions. 8) There is no mention of tDOM stoichiometry. How do DON and DOP discharged by the rivers affect primary production in the investigated area? Is the simulated primary production more realistic when riverine discharge was considered in the model? This question could be answered by comparing the model experiment with tDOM with the experiment without tDOM

---

## Referee Comment (RC2) · Anonymous Referee #2 · 20 Sep 2018

The manuscript by Fransner et al. presents an analysis of pCO2 data from the Gulf of Bothnia together with a biogeochemical model of the basin. Several model scenarios are presented that make different assumptions about remineralisation of tDOC. The authors thereby show that the high pCO2 values in the observational data are only consistent with a model in which tDOC is remineralised rapidly by microbial processes, and in which tDOC also increases light attenuation and thereby reduces primary production close to the coast.

In my opinion, this manuscript presents an insightful analysis that helps us understand the important question of the fate of tDOC in the sea. The manuscript is well written,

clearly structured, and the data are presented clearly. While I am no expert in biogeo-chemical modelling, their model scenarios seem to me to be appropriate for testing their hypotheses, and I believe that their conclusions are justified by the results.

I therefore only have minor questions and comments for the authors to consider. These are as follows:

1. I think it would be helpful to have a map of surface salinity, either seasonally resolved or as a monthly climatology. This would help the reader to link the maps and the scatter plots of pCO2 against salinity. This could even be in the supplementary material.

2. What is the data source for the riverine carbon and other chemical fluxes? I presume that the river runoff from EHYPE refers to the freshwater flux rather than the chemical fluxes, or did I misunderstand that?

3. I agree that Figs 4 and 5 show that the 1Y model comes closest to reproducing the pCO2 observations. However, it seems to me that there are quite a lot of very high pCO2 data that are not predicted by any of the models (esp. in Mar, Apr, and May). Could you maybe add some discussion, even if speculative, about what might be causing even higher pCO2 than in the model?

4. In Section 3.3, I see what you mean by the 10Y remineralisation rate in Fig 6 showing a more spread-out pattern than 1Y. However, in Fig 5, the lines of 10Y and 1Y are almost identical, except below salinity 3 in Jan–May. Why is there no clearer impact on the pCO2?

5. I'm less convinced of the estimates of remineralisation time-scales that the authors calculate on page 8. They are using a simple exponential decay model that assumes that the entire tDOC pool is potentially labile, and then take the concentration reported for the final time-point in each incubation to calculate the time-scale. I've not had time to look through the references myself, but degradation experiments like these typically take measurements at multiple time-points. I think the authors should really confirm

by checking the cited papers again that a single decay model without an asymptote really is justified for each case, as opposed to a more complicated exponential decay model in which one fraction is labile and one fraction is refractory. Maybe the original data from these incubations could even be re-plotted as a supplementary figure with the present authors' decay model superimposed. If the original data do not agree well with the exponential model proposed here, then the authors should discuss possible reasons why microbial remineralisation might be more active in the environment than seen in incubations (maybe priming? Differences in microbial community?).

6. Page 8 bottom line: the units are incomplete for the CO2 uptake rate, and in both cases it should read "m-2" instead of "m2".

7. Page 9 line 10: I got confused here when the authors refer to "CO2 uptake" in the Bothnian Bay, since they say before that the entire Bothnian Bay is a CO2 source to the atmosphere. This needs either correction or better explanation.

---

## Author Response (AR1)

Dear Dr. Ciavatta,

We are pleased to submit a revised version of "Remineralization rates of terrestrial DOC as inferred from CO2 supersaturated waters". In this version we have taken into account all the comments by the reviewers. We have in particular clarified and discussed our choice of a simple decay model, and clarified equation one and the definition of labile tDOC. We have also included figure S5 in the manscript, as you suggested.

We want to thank you and the two reviewers for the work you have done, which have significantly improved the manuscript and made it much clearer.

Please find below point by point answers to the reviewers comments.

Sincerely,
Filippa Fransner and co-authors

**Response to anonymous referee #1**

First of all we want to thank referee #1 for his/her comments that are very constructive and will help improving the manuscript!
Below you will find a response to each comment. The referee's comments are marked in bold and our responses are to be found just below.
The changes we have made in the manuscript are marked in blue italic.

**1) The authors use a quite complex biogeochemical model (BFM) which describes the planktonic ecosystem through a numbers of different plankton functional types. The latter include explicit bacteria and two species of DOM (labile and semi-labile). However, when considering tDOC, the authors use a simplistic decay function assuming that tDOC is all consumed in 1 or 10 years. Why tDOC was not assumed to be cycled by bacteria which are already modeled within the BFM? By using a fixed decay constant, remineralised tDOC goes directly into the DIC pool which is a simplification. Indeed the bacterial growth efficiency in estuaries and coastal zone is relatively high (del Giorgio and Cole 1998) implying that a substantial fraction of DOC assimilated by bacteria is incorporated into bacterial biomass. This might affect the ecosystem in various ways e.g. by affecting grazing (HNAN), the competition between bacteria and phytoplankton for nutrients and the production of recalcitrant DOC. It is also strange that the authors seem to use a different approach for the riverine POC which is assumed to be used by bacteria in the same way as marine POC. I think that the different approach-i.e. the lack of explicit bacterial utilization- used for the experiment with tDOC (the one leading to the main result of the paper) needs to be discussed and justified .**

The referee is right that we haven't explained clear enough the reason behind our approach, and we will make this clearer in the manuscript as described below:

The reason to why we have chosen to use a simplistic decay function for tDOC was firstly to be consistent with the Fransner et al. 2016 paper, on which this study is based. In this paper the decay rate of tDOC is investigated and compared with actual estimates of tDOC concentrations in the estuary, and it is found that the decay can be modeled by either using a rate of 1 year on 80% of the tDOC, or 10 years on 100% of the tDOC. The idea of this study is primarily to investigate which of these rates (if any) that are more realistic when comparing modeled pCO2 to the observed one. In other words the main aim of this paper is to answer the question "Is remineralization (by bacteria and/or sunlight) an important removal pathway of tDOC in the Gulf of Bothnia, and in that case, on what time scale does it occur". The discussion that follows about the underlying process (bacterial/photo-remineralization) is more of a secondary result.

*This has been clarified in the introduction (page 2, lines 29-33), and in section 2.3 (page 4 lines 29-34, and page 5, lines 1-3).*

We agree that it would be interesting to actually let the bacteria degrade the tDOC and to investigate how this affects the competition with phytoplankton. However, we think that when doing so the model output should also be compared to bacterial biomass and growth rates, and that would be enough material for a paper on its own. We therefore think that this is out of the scope of this paper and that it would be an interesting follow up paper.

*We have added a paragraph on "Future studies" (section 4.4) where we discuss this.*

The reason to why the POC is utilized by the bacteria is that it is a built-in feature in the BFM model (the terrestrial DOC was added by ourselves). As the tPOC concentrations are very low compared to the tDOC concentrations, it doesn't have significant impact on our results.

*This has been clarified on page 4, lines 24-27.*

**2) Equation 1(1).**
**This equation is not very clear to me: If Kdtdoc represents the contribution of tDOC to the total light extinction, it should have the same units as the total Kd (i.e. m-1, as presented in Fig 3). The units reported at line 30 of page 4 seem to refer to the specific adsorption coefficient (see equation 9 in Vichi et al 2007) which (I guess) is represented by the parameter '1.0' multiplied by tDOC in eq 1.**

The referee is correct, here we have made a mistake. Kdtdoc represents the contribution of tDOC to the total light extinction and should have the units (m-1).

*This has been corrected (page 5, line 18).*

**3) tDOC is given in ug m-3 which is quite unusual for marine DOC (usually given in mmol m-3) this of course is not a big problem but from eq 1, tDOC concentrations seem to be very low (assuming a max value of Kdtdoc of 7.5). What is the concentration of tDOC given as input to the model? And what is the concentration of the simulated total DOC?**

Also here we have made a mistake; the labile tDOC concentrations amounts to 7500 mg m-3 in the model. So equation number 1 should be written:

$$k_{d_{tDOC}} = 0.15 + 10^{-3} tDOC$$

*Equation 1 has been corrected.*

**4) Why Kdtdoc is not equal to 0 when tDOC is zero?**

This is to take into account the contribution of the refractory fraction of tDOC to the light extinction coefficient, which is not modeled explicitly in the 1Y and the 1YS experiments (see also your comment number 6).

*This has been clarified at page 5, lines 19-21.*

**5) The authors cited different papers reporting different light extinction coefficients (differing by more than one order of magnitude). This suggests that the parameters used to simulate kd are very uncertain. I think that a sensitivity analyses would be useful to understand how the presented results (relative to the exp. 1YS) are affected by the choice of the specific light absorption parameters.**

When investigating this we also did simulations where the light extinction coefficient didn't reach as high as 7.5 (and reached values about one order of magnitude smaller, as in the cited papers). With this the modeled pCO2 drawdown was still too high compared to observations in the low salinity-region. The aim of this experiment was only to provide a possible

explanation to the high pCO2 values also during the productive season. More research (and simultaneous measurements of DOC and PAR) are needed to get a deeper insight on the effects of tDOC on the light extinction and primary production.

*We have added a discussion on this on page 10 (lines 9-14).*

**6) Only the labile fraction of tDOC is assumed to contribute to light extinction. However the biologically refractory fraction of tDOC can be composed by aromatic compounds which strongly interact with light (e.g Stubbins et al. 2010, L&O )**

See answer to your comment number 4.

**7) No mention of the model skills in reproducing broad ecosystem variables (apart from DIN, DIP and pCO2) and fluxes. For example, is the primary production simulated in the various experiments realistic? Are there data available for comparison? If not, simulated values of Chl and primary production could be at least discussed in the context of what is observed in similar areas/ecosystems. I appreciate that the authors refer the reader to a previous paper for a complete validation of the model, however, it would be nice to see a summary of that validation in this manuscript. Additionally it would be very useful to see how the model performance varies in the different experiments reported here. For example, is chl and primary production simulated in exp 1yS more realistic than in the other scenarios investigated?. Without such (at least qualitatively) comparison the reader remains uncertain about the robustness of the conclusions.**

*The referee is right that it could be useful to show, and we have therefore added a validation similar to the one in Fransner et al. 2018 in the supplementary material (figures S6-S19).*

The differences between the experiments presented here and the one in Fransner et al 2018 are minor for the monitoring stations located in the middle of the basins. The largest differences are found in the coastal areas (which is why we show figure S4 in the supplementary material). At these stations there are not enough measurements of chlorophyll to make a validation of the model, which is why we only show DIN and DIP.

**8) There is no mention of tDOM stoichiometry. How do DON and DOP discharged by the rivers affect primary production in the investigated area? Is the simulated primary production more realistic when riverine discharge was considered in the model? This question could be answered by comparing the model experiment with tDOM with the experiment without tDOM.**

In all experiments there is a release of terrestrial organic nutrients (DOM and DOP) from the rivers. Its release and degradation is kept constant over all experiment to make sure that differences in pCO2 are not caused by changes in primary production. This is explained at line 20 in the manuscript. Earlier studies and sensitivity experiments have shown that organic nutrients are important nutrient sources for phytoplankton in the Baltic Sea.

*We have added some text on page 5 (lines 7-10).*

**References:**
Fransner, F., Nycander, J., Mörth, C.-M., Humborg, C., Meier, M. H. E., Hordoir, R., Gustafsson, E., and Deutsch, B.: Tracing terrestrial DOC in the Baltic Sea—A 3-D model study, Global Biogeochemical Cycles, 30, 134–148, https://doi.org/10.1002/2014GB005078, https://agupubs. onlinelibrary.wiley.com/doi/abs/10.1002/2014GB005078, 2016.

Fransner, F., Gustafsson, E., Tedesco, L., Vichi,M., Hordoir, R., Roquet, F., Spilling, K., Kuznetsov, I., Eilola, K.,Mörth, C.-M., Humborg, C., and Nycander, J.: Non-Redfieldian Dynamics Explain Seasonal pCO2 Drawdown in the Gulf of Bothnia, Journal of Geophysical Research: Oceans, 123, 166–188, https://doi.org/10.1002/2017JC013019, https://agupubs.onlinelibrary.wiley.com/doi/abs/10.1002/2017JC013019, 2018.

**Response to anonymous referee #2**

We want to thank referee #2 for his/her work and the helpful comments that has improved our manuscript. Below you find a response to each comment. The referee's comments are marked in bold and our responses are to be found just below.

The changes we have made in the manuscript are marked in blue italic.

**1. I think it would be helpful to have a map of surface salinity, either seasonally resolved or as a monthly climatology. This would help the reader to link the maps and the scatter plots of pCO2 against salinity. This could even be in the supplementary material.**

*This is a good idea, we have added it in the supplementary material (figure S2).*

**2. What is the data source for the riverine carbon and other chemical fluxes? I presume that the river runoff from EHYPE refers to the freshwater flux rather than the chemical fluxes, or did I misunderstand that?**

Indeed, the river runoff from EHYPE only contains freshwater fluxes. The chemical fluxes have been calculated from measurements of river concentrations together with the freshwater fluxes from EHYPE as in Fransner et al 2018.

*We have clarified this on page 3, lines 22-23.*

**3. I agree that Figs 4 and 5 show that the 1Y model comes closest to reproducing the pCO2 observations. However, it seems to me that there are quite a lot of very high pCO2 data that are not predicted by any of the models (esp. in Mar, Apr, and May). Could you maybe add some discussion, even if speculative, about what might be causing even higher pCO2 than in the model?**

We agree that this should be discussed in deeper detail and will therefore add it in the manuscript. One explanation can be that we have a relatively simple degradation model assuming that the tDOC only consists of two pools of different lability. In reality there could for example be one additional pool that is degraded with a faster rate than we use and that is consequently quickly removed in the low salinity region, with a larger impact on the pCO2.

*We have clarified this on page 7 (lines 121-1) and added a discussion on page 8 (18-23).*

**4. In Section 3.3, I see what you mean by the 10Y remineralisation rate in Fig 6 showing a more spread-out pattern than 1Y. However, in Fig 5, the lines of 10Y and 1Y are almost identical, except below salinity 3 in Jan–May. Why is there no clearer impact on the pCO2?**

This is a very good remark! It is true that the pattern of the remineralization rate and the pCO2 difference (related to the 10Y experiment) in Figure 6 is not very identical. If you compare Figure 6b to the bathymetry of the domain (Figure 1b) you will see that that the column-integrated remineralization rate is higher in the deeper parts of the domain. This means that the remineralization of the terrestrial DOC is more spread out over the whole water column, and that there is also remineralization taking place below the thermocline/halocline, which does not directly impact the surface water pCO2. We will add a description of this in the manuscript. We also see that we haven't explained that the difference in the pCO2 only refers to the surface water pCO2, and we will correct this.

*We realized that it would be more appropriate to show a map of the remineralization per volume unit, we therefore decided to change figure 6 a and b. This delivers our message in a clearer way than when showing remineralization per area unit. We do no longer need to discuss the effect of the bathymetry on the remineralization rate per area unit.*

**5. I'm less convinced of the estimates of remineralisation time-scales that the authors calculate on page 8. They are using a simple exponential decay model that assumes that the entire tDOC pool is potentially labile, and then take the concentration reported for the final time-point in each incubation to calculate the time-scale. I've not had time to look through the references myself, but degradation experiments like these typically take measurements at multiple time-points. I think the authors should really confirm by checking the cited papers again that a single decay model without an asymptote really is justified for each case, as opposed to a more complicated exponential decay model in which one fraction is labile and one fraction is refractory. Maybe the original data from these incubations could even be re-plotted as a supplementary figure with the present authors' decay model superimposed. If the original data do not agree well with the exponential model proposed here, then the authors should discuss possible reasons why microbial remineralisation might be more active in the environment than seen in incubations (maybe priming? Differences in microbial community?).**

We agree that our model of remineralization is very simple (which could partly be an explanation to your question number 3) and that there are more sophisticated models that can resolve different pools of DOC with different lability. Indeed, some of the references that we show in the table show a time series of a relative change or actual concentrations (Herlemann et al. 2014, Asmala et al 2014, Hulatt et al. 2014). It would be interesting to plot everything in one figure, but we believe that extracting the concentrations from the figures with several sampling points (in time) would be too difficult (the DOC concentration-axis is not highly resolved enough to do it by eye).
However, we agree that this deserves a discussion, which we didn't have in the first round of our manuscript. We will therefore add a discussion on different remineralization models and the uncertainties associated with our model, in comparison to the remineralization experiments.

*We have added this discussion on page 9 (lines 18-29) and 10 (lines 1-3) and made some minor changes in the choices of words on page 9, to make our intention with this equation clearer.*

**6. Page 8 bottom line: the units are incomplete for the CO2 uptake rate, and in both cases it should read "m-2" instead of "m2".**

*Thanks, these have been corrected.*

**7. Page 9 line 10: I got confused here when the authors refer to "CO2 uptake" in the Bothnian Bay, since they say before that the entire Bothnian Bay is a CO2 source to the atmosphere. This needs either correction or better explanation.**

*We have changed this to outgassing, and in the figure/table we only write air-sea CO2 exchange, where negative values indicate an outgassing, and positive values indicate an uptake.*

**References:**
Fransner, F., Gustafsson, E., Tedesco, L., Vichi,M., Hordoir, R., Roquet, F., Spilling, K., Kuznetsov, I., Eilola, K.,Mörth, C.-M., Humborg, C., and Nycander, J.: Non-Redfieldian Dynamics Explain Seasonal pCO2 Drawdown in the Gulf of Bothnia, Journal of Geophysical Research: Oceans, 123, 166–188,

https://doi.org/10.1002/2017JC013019, https://agupubs.onlinelibrary.wiley.com/doi/abs/10.1002/2017JC013019, 2018.

---

## Author Response (AR3)

Dear Dr. Ciavatta,

We are pleased to submit a revised version of "Remineralization rate of terrestrial DOC as inferred from CO2 supersaturated waters".
We have taken into account the comments from reviewer 1 as stated out in the point by point answers on the next page. Please also find attached a manuscript with marked up differences between the new manuscript and the previous one.

We want to express our gratitude to you and the two reviewers for the work you have done.

Sincerely,
Filippa Fransner and co-authors

**Answers to reviewer 1**

**I am generally satisfied by the way the authors addressed my comments.**
**I only have a few additional points which the authors could consider before publication:**

*We want to thank reviewer 1 for rereading our manuscript and for these additional comments. Please find below our answers and a description of the modifications we have done in the text.*

**1) While eq 1 is now better described, the meaning (and the units) of the factor 10-3 are still not described in the text**

*We added a description on page 5, line 18 in the new manuscript.*

**2) Page 5, Lines 8-10: "in All experiments the terrestrial derived organic nutrients are subject to a degradation..". Is this really the case for all the experiments (I guess not) or only for the TP, 1Y and 10Y? I think it would be better to specify to avoid confusion**

*This is the case for all experiments. We have clarified this on lines page 5, 8-10 in the revised manuscript.*

**3) I would suggest to introduce a bit better the function f(sal) reported in table 1**

*We have specified the number of the equation in Fransner et al. 2018, where it can be found and introduced it better in the table caption. (page 5, line 12 and Table 1 in the new manuscript).*

**4) It would be nice to see how the simulation BIO performs with respect to 1Y and 1YS when compared with observations (nutrients and chlorophyll).. This comparison could provide additional support to the idea that the inclusion of riverine DOM is crucial to properly simulate the investigated ecosystem**

*We see your point, and it could have been interesting if the experiments would have been designed in another way. However, our experiments are designed to focus on the carbon in the DOM, and only difference between BIO and 1Y experiment is the inclusion of terrestrial organic carbon. The input and degradation of organic nutrients is constant over the experiments (to avoid changes in the primary production). Therefore, the only difference between BIO and 1Y will be in the carbonate system, and there is no effect on the nutrients and chlorophyll.*

[revised manuscript text omitted]